# Event-Adaptive State Transition and Gated Fusion for RGB-Event Object Tracking

## Abstract

Existing Vision Mamba-based RGB-Event (RGBE) tracking methods suffer from using static state transition matrices, which fail to adapt to variations in event sparsity. This rigidity leads to imbalanced modeling—underfitting sparse event streams and overfitting dense ones—thus degrading cross-modal fusion robustness. To address these limitations, we propose MambaTrack, a multimodal and efficient tracking framework built upon a Dynamic State Space Model (DSSM). Our contributions are twofold. First, we introduce an event-adaptive state transition mechanism that dynamically modulates the state transition matrix based on event stream density. A learnable scalar governs the state evolution rate, enabling differentiated modeling of sparse and dense event flows. Second, we develop a Gated Projection Fusion (GPF) module for robust cross-modal integration. This module projects RGB features into the event feature space and generates adaptive gates from event density and RGB confidence scores. These gates precisely control the fusion intensity, suppressing noise while preserving complementary information. Experiments show that MambaTrack achieves state-of-the-art performance on the FE108 and FELT datasets. Its lightweight design suggests potential for real-time embedded deployment.

## 1 Introduction

As a core task in computer vision, visual object tracking holds irreplaceable application value in autonomous driving, robotic navigation, human-computer interaction, and other domains (Marvasti-Zadeh et al., 2021; Qiao et al., 2023). Its primary objective is to continuously localize specific targets in video streams, which requires not only precise capture of appearance features (e.g., texture, shape) but also reliable motion trajectory prediction in complex dynamic environments. However, traditional RGB camera-based tracking systems face significant challenges in real-world scenarios: First, the fixed sampling rate of RGB frames (typically 30–60 FPS) leads to motion blur for high-speed targets, especially in scenarios such as drone diving or vehicle sharp turns. Feature extraction errors in blurred regions accumulate over time, ultimately causing tracking drift. Second, under low-light or sudden illumination changes, the limited dynamic range of RGB cameras (approximately 60–70 dB) results in overexposure or underexposure, leading to loss of target appearance information (Maqueda et al., 2018; Alismail et al., 2016). Finally, the dense sampling mechanism of RGB data generates substantial computational redundancy in scenarios with static backgrounds and subtle target movements, hindering energy-efficient embedded deployment.

The bio-inspired perception mechanism of event cameras offers a novel solution to these challenges (Wang et al., 2023b; Ebadi et al., 2023). By asynchronously outputting pixel-level brightness changes (event streams), event cameras exhibit microsecond-level temporal resolution, a high dynamic range ($> 140dB$), and low power consumption, enabling effective capture of visual dynamics in high-speed motion and extreme lighting conditions. For instance, when a target suddenly accelerates, an event camera can generate hundreds of event pulses within the interval of a conventional RGB frame, providing sub-millisecond motion cues for the tracker (Zhu et al., 2019; Gehrig et al., 2019; Scheerlinck et al., 2018; Liang et al., 2023). However, the sparsity and unstructured nature of event streams introduce new challenges: On one hand, event data lack absolute brightness information, making it difficult to reconstruct holistic target features (e.g., color, texture) independently. On the other hand, existing frame-based deep learning models struggle to process asynchronous event streams directly, necessitating specialized spatiotemporal representation methods. Thus, fusing the

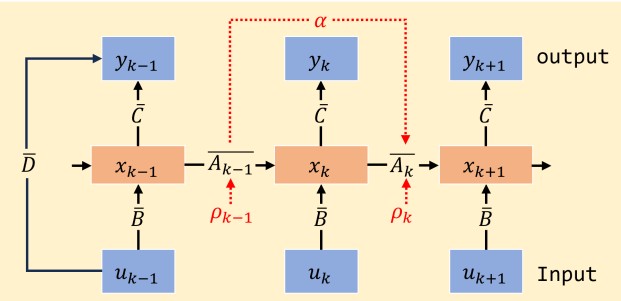

Figure 1: Illustration of the proposed DSSM module with the event-adaptive state transition mechanism. Based on the standard State Space Model (SSM), our DSSM module incorporates an event-adaptive state transition mechanism (highlighted in red), which dynamically modulates the transition dynamics according to the event stream, enabling better temporal modeling under varying event sparsity.

global semantic information from RGB modalities with the high-frequency dynamic responses from event modalities becomes imperative to enhance the robustness and adaptability of tracking systems.

Building on this idea, a variety of RGB-Event (RGBE) fusion-based object tracking methods (Zhang et al., 2023; Zhu et al., 2023b; Zhang et al., 2021; Wang et al., 2024b) have emerged in recent years, aiming to combine the rich texture information of RGB images with the high dynamic responsiveness of event streams to tackle visual challenges in complex environments. Representative works such as AFNet (Zhang et al., 2023), VisEvent (Wang et al., 2023a), and CEUTrack (Tang et al., 2022) have adopted strategies like multimodal alignment, cross-modal fusion, and masked modeling to effectively enhance tracking robustness and adaptability.

Despite the remarkable progress achieved by Transformer-based architectures in RGBE tracking, their inherent structural limitations remain non-negligible: the Transformer (Khan et al., 2022; Liu et al., 2021; Gehrig & Scaramuzza, 2023) architecture faces several critical limitations: its self-attention mechanism suffers from quadratic computational complexity, making it inefficient for modeling long sequences; during training and inference, its heavy reliance on key-value caching results in significant memory consumption and latency overhead; more fundamentally, it lacks the ability to model the sparsity variations inherent in event streams, making it difficult to dynamically adjust the modeling granularity under varying event densities, thereby limiting its spatiotemporal adaptability in RGBE tracking tasks.

Recently, the introduction of the Mamba (Gu & Dao, 2023) architecture has significantly alleviated the computational and memory bottlenecks of Transformers. Meanwhile, the event modality shows unique advantages in providing fine-grained motion cues and localized high-dynamic details, which are crucial for addressing core tracking challenges such as lighting variations and motion blur. However, directly adopting the static state transition model of Mamba fails to effectively extract and utilize the sparse spatiotemporal characteristics of event data. Therefore, it is essential to develop a dedicated strategy for dynamic state modeling and cross-modal fusion tailored to event streams.

To address this challenge, we propose MambaTrack, a lightweight multimodal object tracking framework based on dynamic state-space modeling, which incorporates two key mechanisms in a unified design. Specifically, we develop an event-adaptive state transition mechanism that uses a learnable scalar $\alpha$ to adaptively modulate the state evolution rate, allowing the model to adjust its temporal modeling granularity according to the density of event pulses. This enables differentiated modeling in both sparse and dense event scenarios, thereby enhancing the representational capacity of the event branch. In parallel, we introduce a Gated Projection Fusion(GPF) module that maps RGB features into the event space via a multi-layer perceptron (MLP) and generates adaptive gating coefficients based on event density and RGB confidence. Meanwhile, it also modulates RGB features using event features in a similar manner, enabling bidirectional cross-modal weighted fusion.

In summary, our contributions are threefold.

- We propose MambaTrack, an efficient multimodal fusion framework tailored for RGB-Event object tracking. It is built upon an event-adaptive state transition mechanism that effectively addresses the modeling challenges caused by event sparsity variations.

- We design a Gated Projection Fusion(GPF) module that adaptively regulates the fusion strength based on the density and confidence of each modality, effectively promoting robust feature interaction and suppressing noise interference.

- We validate the effectiveness of the proposed MambaTrack through comprehensive experiments on two representative RGB-Event tracking benchmarks, FELT and FE108, where it consistently achieves competitive performance, demonstrating its robustness and generalization capability.

## 2 RELATED WORK

### 2.1 MAMBA

The State Space Model (SSM) originally served as a mathematical framework for characterizing the dynamics of dynamic systems. S4 (Gu et al., 2021) reparameterizes the structured state matrix by decomposing it into a combination of low-rank and normal terms and computes truncated generating functions in the frequency domain, reducing computational complexity to $\tilde{O}(N + L)$ and significantly improving long-sequence modeling efficiency. S5 (Smith et al., 2022) further extends the traditional single-input single-output (SISO) SSM to a multi-input multi-output (MIMO) structure, achieving more efficient parallel scanning computations through parameterized diagonalized dynamic matrices. Building on this foundation, Mamba (S6) (Gu & Dao, 2023) introduces a data-dependent SSM layer and parallel scanning selection mechanism, which not only greatly enhances inference speed but also demonstrates superior performance over equivalently scaled Transformers in vision tasks .

In the field of computer vision, Vision Mamba (Zhu et al., 2024) first integrates Mamba into a generic vision backbone architecture, enabling efficient modeling of image sequences through bidirectional Mamba blocks and positional embeddings; VMamba (Liu et al., 2024) proposes a cross-scan module (CSM) to convert non-causal visual images into ordered patch sequences through spatial domain traversal, significantly enhancing long-range dependency modeling; further, EfficientVMamba (Pei et al., 2025) reduces computational overhead while maintaining performance through lightweight design and dynamic pruning strategies; in multimodal perception tasks, MFNet (Hong et al., 2025) constructs a multipath feature fusion framework using Mamba's parallel scanning mechanism, effectively resolving temporal alignment issues between RGB and event data.

In other domains, S4ND (Nguyen et al., 2022) extends SSM's continuous signal modeling capabilities to multidimensional data , achieving spatiotemporal joint modeling through high-dimensional polynomial projections; Pan-Mamba (He et al., 2025) introduces channel swapping and cross-modal modules, establishing a new paradigm for efficient interaction and fusion of multimodal information; notably, FusionMamba (Dong et al., 2024) incorporates an adaptive weighting mechanism in cross-modal tasks, enabling Mamba to dynamically balance the representation capabilities of different modalities. These works collectively demonstrate Mamba's flexibility and generalization capabilities in complex data modeling, laying a foundation for its broader applications across diverse fields.

### 2.2 RGB-EVENT TRACKING

Research on RGB-Event (RGBE) fusion for object tracking has gained increasing attention, aiming to combine the rich texture of RGB images with the high dynamic response of event streams to address visual challenges in complex scenarios. Zhang et al. (2021) introduces a multi-modal alignment and fusion module to effectively integrate RGB and event data with different sampling rates, achieving robust tracking at high frame rates. The VisEvent (Wang et al., 2023a) method constructs a comprehensive dataset containing 820 visible-event video pairs and establishes a baseline using a cross-modality Transformer (CMT) to enhance feature interaction. CEUTrack (Tang et al., 2022) unifies RGB frames and color-event voxels through a single-stage backbone, enabling simultaneous feature extraction, matching, and interactive learning. AFNet (Zhang et al., 2023) further incorporates an event-guided cross-modal alignment (ECA) module and cross-correlation fusion (CF) to im-

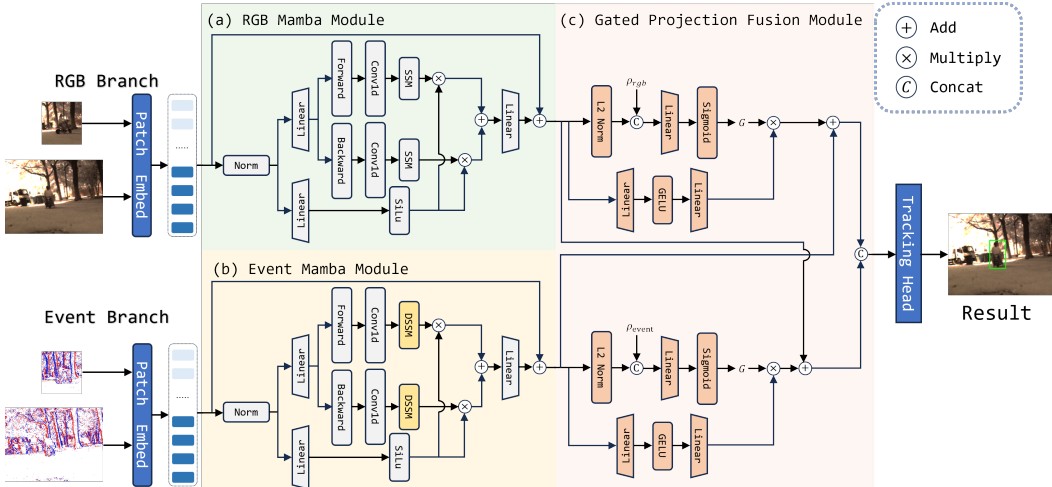

Figure 2: Overall architecture of our proposed MambaTrack. The framework consists of two main components: feature extraction and feature fusion. The feature extraction branch includes (a) the RGB Mamba Module and (b) the Event Mamba Module, which are designed to extract modality-specific features. These features are then adaptively integrated in (c) the Gated Projection Fusion (GPF) Module to enhance cross-modal representation.

prove target localization in dynamic environments. Zhu et al. (2023b) proposed a masked modeling strategy that randomly masks tokens of modalities to bridge the distribution gap between RGB and event data, thereby enhancing model generalization. HDETrack (Wang et al., 2024b) pioneers the application of knowledge distillation in multimodal tracking, transferring multi-view (event image-voxel) knowledge to single-modality event tracking and expanding the utility boundaries of event data.

As Transformer-based approaches have dominated this field, recent studies have begun exploring more efficient and lightweight alternatives. The emergence of the Mamba architecture offers a promising direction for RGB-E tracking due to its simplicity and computational efficiency. Mamba-FETrack (Huang et al., 2024), following this trend, adopts the Mamba backbone to enable adaptive RGB-E fusion by integrating dual-modality gating signals and leveraging Mamba's temporal modeling capabilities. Our approach further explores the potential of Mamba in RGB-E object tracking, demonstrating its effectiveness in balancing accuracy and efficiency in complex scenarios.

## 3 OUR PROPOSED APPROACH

In this paper, we propose MambaTrack, as shown in Fig. 2, a novel RGB-Event bimodal object tracking framework based on an event-adaptive state transition mechanism and Gated Projection Fusion(GPF) module. Our model first temporally aligns asynchronous event streams to RGB frame timestamps via linear interpolation, generating spatiotemporal voxel grids. RGB frames are processed through patch embedding followed by spatiotemporal positional encoding. Subsequently, the event stream and RGB frames are fed into Dynamic State Space Model(DSSM) and static SSM (Gu & Dao, 2023) branches, respectively, to extract modality-specific features. We then design a lightweight MLP to project RGB features into the event feature space and dynamically regulate residual fusion intensity using gating weights , effectively suppressing interference from noisy modalities. Finally, the fused features are directly fed into the tracking head for target localization, and the entire framework is trained end-to-end using a joint classification-regression loss.

### 3.1 INPUT REPRESENTATION

Our framework adopts RGB video frames and asynchronous event streams as dual-modal input, achieving spatiotemporal consistency across modalities through dynamic temporal alignment and structured encoding.

For RGB frame,the RGB video sequence is denoted as $\mathcal{I} = \{I_1, I_2, \ldots, I_N\}$ where $I_i$ represents the i-th RGB frame, and N is the total number of frames. We crop the template patch $Z^I$ (target initialization region) and search patch $X^I$ (candidate search region) from RGB frames.

For event stream, the asynchronous event stream is represented as $\mathcal{E} = \{e_j = (x_j, y_j, t_j, p_j)\}_{j=1}^M$,where each event $e_j$ contains pixel coordinates $(x_j, y_j)$,timestamp $t_j$ and polarity $p_j \in \{+1, -1\}$,with M being the total number of events. To achieve temporal alignment with RGB frames, we adopt the time surface representation (Lagorce et al., 2016) of events. For each RGB frame timestamp $t_{\text{RGB}}^{(i)}$,we construct an event window covering its exposure duration and compute event contributions via linear interpolation:

$$V_t(x, y) = \sum_j p_j \cdot \max\left(0, 1 - \frac{|t_{\text{RGB}}^{(i)} - t_j|}{\Delta t}\right) \tag{1}$$

where $\Delta t$ is the exposure time of RGB frames, the weight linearly decays with the temporal distance between events and RGB timestamps.

## 3.2 Modality-Specific Mamba Backbone

Inspired by the hierarchical design of vision Mamba (Zhu et al., 2024), we propose dual independent Mamba branches to extract modality-specific features from RGB and event streams, preserving their inherent characteristics. As shown in Fig. 2, the backbone comprises two parallel branches.

**Vision Mamba Model.** For the frame branch, the vision Mamba model processes the concatenated template and the search token sequence $\mathcal{H}_0^{\text{RGB}} \in \mathbb{R}^{T \times H \times W \times D}$. The input first undergoes layer normalization followed by separate linear projections to generate latent representations $z$ and $x$. The $x$ tensor is then bidirectionally processed by 1D convolution in depth and SiLU activation to produce enhanced features $x'$.

Subsequent processing employs a State Space Model (SSM) where $x'$ is linearly projected to obtain parameter matrices $B$, $C$, and the temporal scaling factor $\Delta$. Bidirectional SSM processes $x'$ through forward and backward directions, with output adaptively gated by projected $z$ via multiplication of SiLU-activated elements (Elfwing et al., 2018) by elements. Final features are obtained by aggregating both directional outputs through summation.

**Event-adaptive State Transition Mechanism.** In RGB-Event tracking, Vision Mamba leverages the linear computational characteristics of SSM to enable more efficient motion feature modeling with fewer parameters while ensuring spatio-temporal alignment accuracy for heterogeneous RGB-Event data. However, the static state transition matrix in Mamba inherently conflicts with the asynchronous spiking nature of event streams. Considering the characteristics of event streams, we designed an event-adaptive state transition mechanism with a Dynamic State Space Model(DSSM), as shown in Fig. 1. The implementation process is as follows:

Given that the dynamic nature of events manifests itself as sparse events during object stasis and dense events during high-speed motion, we introduce a parameter representing event density $\rho_t$, which quantifies the spatial concentration of events within a specific time window by normalizing the number of events to the unit area of the space. The specific computational process is described as follows:

$$\rho_t = \frac{\text{count} V_t}{H \times W} \tag{2}$$

where $\text{count} V_t$ denotes the number of events. Since the event density is a scalar, a trainable matrix $W_a$ is introduced to project the raw input $\rho_t$ into a latent space compatible with the state space model, achieving parameter dynamization. The sigmoid function ensures normalization, preventing numerical instability caused by event stream spikes.The calculation process of the dynamic scaling factor is expressed by the following equation:

$$\beta = \sigma(W_a \rho_t) \tag{3}$$

where $\sigma$ denotes the Sigmoid activation function. Next, we introduce a static prior matrix $A_{base}$ as a reference matrix for the dynamic adjustment factor $\beta$, avoiding drastic fluctuations in the parameters in the initial stage. To adapt to changes in motion states and suppress abrupt changes caused by pulse

spikes in the event stream, ensuring gradual updates of the state transition matrix, we introduce a learnable scalar $\alpha$. The current-state transition matrix $A_t$ is obtained by weighted fusion of current observations and historical states $A_{t-1}$. The specific calculation process is as follows:

$$A_t = \alpha \cdot \beta \cdot A_{\text{base}} + (1 - \alpha) \cdot A_{t-1} \tag{4}$$

where $A_{\text{banse}} \in \mathbb{R}^{D \times D}$ denotes an identity matrix, $\sigma$ denotes the Sigmoid activation function, $\alpha \in (0, 1)$ denotes a learnable scalar.

This design leverages an event-driven dynamic modeling mechanism, which effectively mitigates abrupt changes caused by event spikes while maintaining the stability of state updates. To ensure compatibility with the original Mamba architecture, we incorporate the dynamically generated state matrix $A_t$ as a learnable bias term to enhance the representation capacity of the original transition matrix $A$. The specific computation is as follows:

$$A_{final} = A_t + A \tag{5}$$

This design allows the model to retain the structural stability of Mamba while introducing temporal awareness and data-driven dynamic adaptation, thereby enhancing the capability to model temporal patterns in heterogeneous RGB-Event data.

### 3.3 GATED PROJECTION FUSION MODULE

To address noise interference and information redundancy in RGB–event modality fusion, as shown in Fig. 2(c), we propose a Gated Projection Fusion(GPF) module: this module first feeds the density and confidence of each modality jointly into a gating network to adaptively generate gating coefficients, then uses these coefficients to perform bidirectional weighted fusion of cross-modal projection, thereby suppressing noise while preserving complementary information.

The implementation pipeline is formally described as follows: First, the RGB modality features are projected into the event stream feature domain through a multilayer perceptron (MLP)-based feature space alignment module, formulated as:

$$\Delta F = W_2 \cdot \text{GELU}(W_1 F_{\text{RGB}} + b_1) + b_2 \tag{6}$$

where $W_1 \in R^{D \times D}$ and $W_2 \in R^{D \times D}$ denote learnable weight matrices of the fully-connected layers, $b1, b2 \in R^D$ are bias terms, and $GELU(\cdot)$ represents the GELU activation function (Hendrycks & Gimpel, 2016).

Next, we introduce a density-aware gated fusion mechanism, which adaptively generates a gating coefficient by jointly leveraging event density and RGB feature confidence. This coefficient is used to regulate the extent to which RGB features complement the event modality, enabling more effective cross-modal feature fusion that preserves complementary information while suppressing redundancy and noise. The gating coefficient is computed as follows:

$$G = \text{Sigmoid}\left(W_g\left[\rho(t); \|F_{\text{RGB}}\|_2\right]\right) \tag{7}$$

where $W_g \in R^{2 \times 1}$ parameterized the gating weights, $\rho(t) \in R$ indicates the event density at timestamp $t$, $\|F_{\text{RGB}}\|_2 \in \mathbb{R}$ quantifies the confidence of RGB features via L2-norm, and $[\cdot]$ denotes vector concatenation.

Finally, the calibrated RGB features are fused with the event stream features through gated weighting, producing cross-modality enhanced representations:

$$F_{\text{fuse} \to E} = F_{\text{Event}} + G \odot \Delta F \tag{8}$$

where $\odot$ signifies element-wise multiplication, $F_{Event} \in R^D$ corresponds to the raw event stream features, and $\Delta F \in R^D$ represents the aligned RGB feature projection.

Symmetrically, by setting $\rho(t)$ to 1 and swapping the input modalities, the enhanced RGB features can be obtained:

$$F_{\text{fuse} \to R} = F_{\text{RGB}} + G' \odot \Delta F' \tag{9}$$

where the computations of $G'$ and $\Delta F'$ follow the same procedure as above, but with event features as the input modality. Finally, the two fused search region features are concatenated:

$$x = [F_{\text{fuse} \to E}; F_{\text{fuse} \to R}] \tag{10}$$

where the fused features $x$ are directly fed into the tracking head for target localization. This module adaptively suppresses noise and balances complementary features through event density–aware gating.

| Method | SR(%) | PR(%) |
|---|---|---|
| TransT (Chen et al., 2021) | 34.6 | 44.3 |
| ATOM (Danelljan et al., 2019) | 36.2 | 45.9 |
| KYS (Bhat et al., 2020) | 33.1 | 42.4 |
| OSTrack-S (Ye et al., 2022) | 40.0 | 50.9 |
| OSTrack (Ye et al., 2022) | 32.5 | 40.3 |
| AFNet (Zhang et al., 2023) | 28.9 | 36.6 |
| ViPT (Zhu et al., 2023a) | 35.7 | 44.1 |
| **Ours** | **42.5** | **54.0** |

Table 1: Tracking results on FELT SOT dataset

### 3.4 HEAD AND LOSS FUNCTION

We employ the tracking head of OSTrack (Ye et al., 2022). The tracking head outputs include the classification score map, the size of the bounding box and the local offset. We employ the focal loss (Lin et al., 2017), the L1 loss (Girshick, 2015), and the GIoU loss (Rezatofighi et al., 2019) during training.The total loss is as follows:

$$L = \lambda_{focal}L_{focal} + \lambda_1 L_1 + \lambda_{GIoU}L_{GIoU} \tag{11}$$

where $\lambda_{focal} = 1.5$, $\lambda_1 = 5$ and $\lambda_{GIoU} = 2$ are the hyperparameters in our experiment.

## 4 EXPERIMENT

### 4.1 EXPERIMENTAL SETTINGS

**Implementation Details.** We implemented the proposed MambaTrack framework using PyTorch and trained it on 2 NVIDIA RTX 4090 GPUs. Specifically, we adopted the AdamW (Loshchilov & Hutter, 2017) optimizer, with the learning rate, batch size, and weight decay set to 0.0004, 48, and 0.0001, respectively. The learning rate scheduling followed a StepLR strategy with a decay rate of 0.1. For the backbone, we employed a lightweight pre-trained Vision Mamba model.

**Evaluation metrics.** In our experiments, we employ three standard evaluation metrics to assess tracking performance: Success Rate (SR), Precision Rate (PR), and Normalized Precision Rate (NPR). SR evaluates the proportion of frames where the Intersection over Union (IoU) between the predicted and ground truth bounding boxes exceeds a threshold, reflecting the tracker's robustness. PR measures the percentage of frames where the center location error is within a fixed pixel distance, indicating the tracker's localization accuracy. NPR further normalizes this center error by the target size, offering a scale-invariant assessment that better handles objects of varying sizes.

**Dataset.**We evaluate the proposed method on two large-scale RGB-Event tracking datasets: FE108 (Zhang et al., 2021) and FELT (Wang et al., 2024a). FE108 is collected using a grayscale DVS346 event camera, containing 76 training and 32 testing videos, covering various indoor challenges such as motion blur and illumination changes.FELT is the largest frame-event long-term tracking dataset to date, consisting of 742 video sequences with 1.59 million RGB frames and event streams, covering 45 object categories and 14 challenge attributes, enabling comprehensive performance evaluation.

### 4.2 COMPARISONS WITH STATE-OF-THE-ARTS

**Results on FELT dataset.**Table 1 summarizes the performance comparison between our proposed MambaTrack and several state-of-the-art trackers on the FELT dataset. As observed, MambaTrack achieves an SR of 42.5% and a PR of 54.0%, indicating strong overall accuracy. When compared with methods like AFNet (Zhang et al., 2023), our tracker delivers better performance while maintaining a more compact model. Notably, in comparison with ViPT (Zhu et al., 2023a), MambaTrack achieves substantial improvements of +6.8% in SR and +9.9% in PR. These results suggest that our method is particularly well-suited for long-sequence scenarios, benefiting from its enhanced temporal modeling and training efficiency.

| Method | SR(%) | PR(%) |
|---|---|---|
| SiamRPN (Li et al., 2018) | 21.8 | 33.5 |
| SiamBAN (Chen et al., 2020) | 22.5 | 37.4 |
| SiamFC++ (Xu et al., 2020) | 23.8 | 39.1 |
| KYS (Bhat et al., 2020) | 26.6 | 41.0 |
| CLNet (Dong et al., 2020) | 34.4 | 55.5 |
| CMT-MDNet (Wang et al., 2023a) | 35.1 | 57.8 |
| ATOM (Danelljan et al., 2019) | 46.5 | 71.3 |
| DiMP (Bhat et al., 2019) | 52.6 | 79.1 |
| CMT-ATOM (Wang et al., 2023a) | **54.3** | 79.4 |
| **Ours** | 52.7 | **81.7** |

Table 2: Tracking results on FE108 dataset

| # | RGB | Event | DSSM | GPF | SR(%) | PR(%) |
|---|---|---|---|---|---|---|
| 1 | × | | | | 33.4 | 41.2 |
| 2 | | × | | | 41.3 | 52.5 |
| 3 | | | × | | 42.1 | 53.2 |
| 4 | | | | × | 41.9 | 53.4 |
| 5 | | | | | **42.5** | **54.0** |

Table 3: Ablation study for important components on FELT SOT dataset. × represents the component is removed.

**Results on FE108 dataset.** We conduct a quantitative evaluation of MambaTrack on the FE108 benchmark and compare it with several state-of-the-art trackers, as reported in Table 2. Using Success Rate (SR) and Precision Rate (PR) as evaluation metrics, other tracking method such as DiMP (Bhat et al., 2019) achieves 52.6% and 79.1% on SR/PR, while MambaTrack improves the performance to 52.7% and 81.7%, respectively. Notably, when compared with representative methods such as CMT-MDNet (Wang et al., 2023a) and ATOM (Danelljan et al., 2019), our model demonstrates a clear performance advantage, achieving a precision improvement of 10.4 percentage points over ATOM. These results highlight the robustness and effectiveness of MambaTrack in handling challenging indoor tracking scenarios involving multimodal input.

### 4.3 ABLATION STUDY

**Impact of Multimodal Input.** To systematically assess the influence of input modality on RGB-Event tracking performance, we conduct an ablation study under three configurations: RGB-only, Event-only, and combined RGB-Event input. As summarized in Table 3#1, #2 and #5, the results indicate that only RGB input achieves 41.3% SR and 52.5% PR, while only Event input yields slightly lower scores of 33.4% and 41.2%, respectively. In contrast, the full model with fused RGB and Event modalities attains 42.5% SR and 54.0% PR, clearly outperforming the unimodal settings. These findings substantiate the complementary characteristics of RGB and event data in capturing spatial structure and temporal dynamics, and demonstrate the robustness and effectiveness of our fusion strategy in cross-modal feature representation.

**Effectiveness of Event-adaptive State Transition Mechanism.** To rigorously evaluate the contribution of the proposed Event-adaptive State Transition Mechanism in multimodal tracking, we conduct an ablation study by removing its core component—the Dynamic State Space Model (DSSM)—from the overall architecture. As reported in Table 3#3 and #5, excluding DSSM leads to a decrease in Success Rate (SR) from 42.5% to 42.1%, and in Precision Rate (PR) from 54.0% to 53.2%. Although the performance degradation appears moderate, the consistent drop across both metrics substantiates the positive impact of DSSM on temporal state modeling. We attribute this improvement to DSSM's ability to dynamically modulate state transitions in response to variations in event density, thereby enhancing the tracker's capacity to cope with abrupt motion changes, state drift, and event sparsity. These findings confirm the effectiveness of the proposed mechanism in promoting temporal stability and adaptive precision in long-term tracking scenarios.

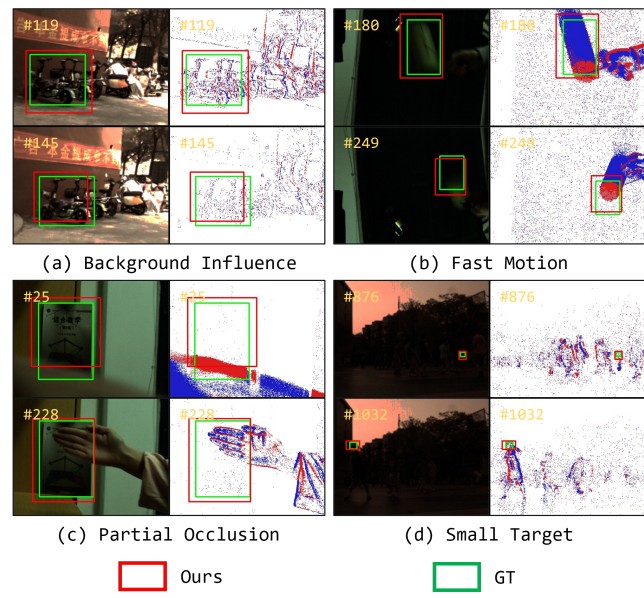

(a) Background Influence          (b) Fast Motion

(c) Partial Occlusion          (d) Small Target

☐ Ours          ☐ GT

Figure 3: Tracking results of our MambaTrack under 4 challenging conditions on FELT SOT dataset.

**Effectiveness of Fusion Module.** To assess the effectiveness of the proposed Gated Projection Fusion(GPF) module in cross-modal integration, we conduct an ablation experiment by removing this component from the full model and comparing the results. As shown in Table 3#4 and #5, the Success Rate (SR) drops from 42.5% to 41.9%, and the Precision Rate (PR) declines from 54.0% to 53.4%, indicating a substantial performance degradation. These results highlight the pivotal role of the GPF module in multimodal fusion. By leveraging modality density and confidence to adaptively control the fusion strength, the module effectively suppresses redundancy and noise while preserving complementary information between modalities, thereby contributing to a notable improvement in overall tracking performance.

### 4.4 VISUALIZATION

To evaluate the effectiveness of our proposed MambaTrack, we present visualizations of its tracking performance under four challenging scenarios: Small Target (ST), Background Influence (BI), Partial Occlusion (POC), and Fast Motion (FM). Fig. 3 compares the predicted bounding boxes with the ground truth annotations. Results show that MambaTrack maintains accurate and robust tracking under all challenging conditions—it can localize small targets, resist background interference, handle partial occlusions, and cope with fast motion. These advantages stem from the event-adaptive state transition mechanism and the Gated Projection Fusion (GPF) module, and the visual results validate its robustness and generalization capacity in real-world scenarios.

## 5 CONCLUSION

This paper proposes MambaTrack, an RGB-Event tracking framework integrating an event-adaptive dynamic state transition mechanism and a gated cross-modal fusion strategy. The Dynamic State Space Model (DSSM) enables adaptive temporal modeling based on event density, while the Gated Projection Fusion (GPF) regulates feature interaction strength via event density and RGB confidence. Extensive experiments on FE108 and FELT show MambaTrack's superior accuracy and robustness. Ablation studies and qualitative visualizations validate each module's effectiveness in enhancing spatiotemporal adaptability and multimodal complementarity; its modular design also makes it promising for future real-time and embedded tracking applications. Future work will explore tracking head designs better suited to event-based data, aiming to improve robustness and generalization in dynamic or complex environments.

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
