# OpenReview forum: "Event-Adaptive State Transition and Gated Fusion for RGB-Event Object Tracking"
_ICLR.cc/2026/Conference — ICLR 2026 Conference Withdrawn Submission_

### Official Review · Reviewer_vc3k · 2025-10-17

**Soundness:** 1
**Presentation:** 2
**Contribution:** 1
**Rating:** 2
**Confidence:** 5

**Summary:**

The paper proposes MambaTrack, an RGB-Event tracking framework that integrates an Dynamic State Space Model (DSSM) and a Gated Projection Fusion GPF) strategy. Specifically, the DSSM enables adaptive temporal modeling based on event density, while the GPF modulates the strength of cross-modal feature interaction according to both event density and the confidence of RGB features.

**Strengths:**

1. The issue focused in this paper, namely the efficient fusion of sparse event-modality data and RGB-modality data, is an important problem in the RGB-Event tracking task.

2. The detailed introduction of the method in Section 3 facilitates the reader’s understanding.

**Weaknesses:**

1. The motivation for modeling in this paper is somewhat confusing. The introduction of the core model design, namely Mamba, is intended to address the issue of high computational cost (typically when dealing with long temporal sequences). However, the model proposed in this paper takes only a few images as input, which makes the justification for introducing Mamba less convincing.

2. The model design in this paper lacks novelty, and the overall contributions are not sufficient. All the module designs in this paper are merely incremental works based on the existing Mamba Module.

3. The baseline models compared in this paper are all early models, and there is a lack of comparison with some of the latest RGB-E trackers, such as CSTrack (ICML 2025) and SUTrack (AAAI 2025). Moreover, this paper lacks evaluation on some representative benchmarks, such as VisEvent.

4. The ablation study analysis in this paper is not sufficient (only the experimental results shown in Table 3 are provided). Additional analyses, such as computational efficiency, comparisons of different backbone network structures (Mamba vs. ViT), and the impact of related hyperparameters, are also needed.

**Questions:**

Please refer the above weaknesses.

---

### Official Review · Reviewer_o92w · 2025-10-22

**Soundness:** 2
**Presentation:** 1
**Contribution:** 2
**Rating:** 2
**Confidence:** 4

**Summary:**

In RGB-Event (RGBE) tracking, Transformer architectures suffer from substantial computational overhead. Conversely, the efficient Mamba architecture utilizes a "static" model, which is incapable of adapting to the core characteristic of highly dynamic variations in event stream sparsity, leading to diminished model robustness.

This paper proposes a lightweight MambaTrack framework that addresses this issue through "Event-Adaptive State Transition" and "Gated Projection Fusion." The former enhances Mamba's static model, enabling it to dynamically adjust its internal modeling methodology based on the event stream's density (i.e., sparse or dense). The latter evaluates the reliability of the RGB (e.g., confidence) and event (e.g., density) modalities, subsequently auto-regulating the fusion weights to suppress noise (such as blurred RGB imagery or sparse event data) while retaining salient information.

**Strengths:**

The illustrations and the written exposition are both highly detailed and comprehensible, accurately articulating the proposed methodology.

**Weaknesses:**

The comparative analysis is outdated; the most recent methods included for comparison were published in 2023. It is recommended to incorporate more contemporary SOTA (State-of-the-Art) trackers.

The experimental evaluation is insufficient, as the algorithm's performance was validated on only two datasets. It is suggested that the two tables be consolidated. Furthermore, the qualitative visualizations are limited; more comprehensive results and comparisons with other trackers should be provided.

The method's applicability is highly restricted. Its exclusive suitability for RGB-Event (RGBE) scenarios significantly curtails its practical utility and generalizability.

As demonstrated in Table 3, the proposed modules (DSSM and GPF) yield only marginal performance gains (approximately 0.5%) in the ablation study. Consequently, the value (or significance) of the proposed method is questionable.

**Questions:**

The manuscript posits that the event modality provides a higher temporal resolution. In light of this, is there a one-to-one correspondence between the RGB frames and the event frames within the datasets utilized? If such a correspondence is absent, what methodology is employed to more effectively utilize the surplus event frames?

---

### Official Review · Reviewer_qLns · 2025-10-30

**Soundness:** 2
**Presentation:** 2
**Contribution:** 2
**Rating:** 4
**Confidence:** 5

**Summary:**

The paper proposes MambaTrack, an RGB–event object tracking framework featuring a Dynamic State Space Model (DSSM) for adaptive temporal modeling based on event density, and a Gated Projection Fusion (GPF) to control cross-modal feature interactions. The experiments on FE108 and FELT datasets demonstrate strong accuracy and robustness. Ablation studies and visualizations confirm the effectiveness of each module, and the modular design suggests promise for real-time and embedded applications.

**Strengths:**

1. The topic of object tracking using events and frames is very important in high-speed or low-light scenarios.

2. The writing is clear and easy to understand.

**Weaknesses:**

1. The proposed event-adaptive state transition mechanism essentially performs scalar weighting based on event density and does not introduce structured dynamic modeling. Compared with existing Mamba variants that already incorporate dynamic mechanisms, the novelty is limited. It is difficult to identify a clear contribution to the field of neuromorphic object tracking.

2. The experiments are not comprehensive. The method is only compared against older Transformer-based or traditional approaches, and does not include comparisons with the latest RGB–event object tracking methods, which reduces the overall convincingness of the results.

3. The paper does not provide a detailed evaluation under challenging conditions, such as low-light, high-speed motion, or severe occlusion, making it difficult to assess the actual benefits of the proposed hybrid framework in challenging scenarios.

**Questions:**

Please response each comment in the weaknesses.

---

### Note · Authors · 2025-11-13

I have read and agree with the venue's withdrawal policy on behalf of myself and my co-authors.